# Humans rationally balance detailed and temporally abstract world models
Ari E. Kahn [1] ✉ & Nathaniel D. Daw [1,2]

How do people model the world's dynamics to guide mental simulation and evaluate choices? One prominent approach, the Successor Representation (SR), takes advantage of temporal abstraction of future states: by aggregating trajectory predictions over multiple timesteps, the brain can avoid the costs of iterative, multi-step mental simulation. Human behavior broadly shows signatures of such temporal abstraction, but finer-grained characterization of individuals' strategies and their dynamic adjustment remains an open question. We developed a task to measure SR usage during dynamic, trial-by-trial learning. Using this approach, we find that participants exhibit a mix of SR and model-based learning strategies that varies across individuals. Further, by dynamically manipulating the task contingencies within-subject to favor or disfavor temporal abstraction, we observe evidence of resource-rational reliance on the SR, which decreases when future states are less predictable. Our work adds to a growing body of research showing that the brain arbitrates between approximate decision strategies. The current study extends these ideas from simple habits into usage of more sophisticated approximate predictive models, and demonstrates that individuals dynamically adapt these in response to the predictability of their environment.

A hallmark of human and animal planning is its flexibility: the ability to plan and replan effectively in the face of novel or changing circumstances and outcomes[1]. This requires not only predicting the immediate outcomes of actions based on experience, but also inferring longer-term payoffs, which often depend on a series of subsequent states and actions[2]. A long-considered solution to this problem is for the brain to simulate outcomes through use a cognitive map or internal model[3]. Reinforcement learning (RL) theories formalize this idea in terms of "model-based" (MB) algorithms[4], which employ a learned model of the short-term consequences of actions in order iteratively to simulate the long-run consequences of candidate actions. Other learning approaches aim to simplify this laborious decision-time computation at the expense of reduced flexibility. Notably, model-free (MF) RL algorithms[5] directly learn and cache the long-run aggregate reward expected for performing an action (the key decision variable), without representing or computing over the step-by-step contingencies. This avoids costly model-based search, but at the expense of sub-optimal behavior in some circumstances: for instance, when distal contingencies or goals change in a way that invalidates the cached aggregates. It has been suggested that the brain employs both methods, and that it judiciously trades them off, balancing their costs and benefits according to situational demands[6–9]. This approach suggests a formal, resource-rational account of both healthy[10] and pathological habits[11–13], as well as helping to characterize changes in their deployment in other situations such as over development and aging[14,15].

The successes of this framework as well as the practical limitations of both extreme algorithms[16] have motivated the study of more nimble methods that strike an intermediate balance between flexibility and computational cost. One important class of approaches simplifies internal models and model-based prediction by leveraging *temporal abstraction* over future states[17]. Prominent among these is the Successor Representation (SR)[18], a planning algorithm that models future states (instead of future rewards) aggregated over multiple timesteps. Like MB RL with a one-step model, such a multi-step world model allows computing or recomputing net payoffs (by combining state predictions with up-to-date per-state rewards) — but without the computational expense of step-by-step simulation[19]. However, the modeling assumptions that simplify prediction (forecasting multiple steps ahead at once) are not always valid: if intermediate contingencies or preferences change in a way that invalidates the SR's summarized world model, it will misvalue actions[20]. Such characteristic slips of action both impose a potential cost on SR usage (offsetting gains due to streamlined computation) and render its use empirically detectable. Indeed, a number of studies of human and animal decision making suggest that the brain uses this or similar temporally abstract predictive representations, at least some of the time[20–27]. However, it remains unclear

[1]Princeton Neuroscience Institute, Princeton University, Princeton, NJ, USA. [2]Department of Psychology, Princeton University, Princeton, NJ, USA.
✉e-mail: arik@princeton.edu

whether the concept of resource-rational trade-offs[28] (e.g. balancing the costs and benefits of SR vs. full MB as appropriate to the circumstances) can be extended to this case. Furthermore, this is in part an empirical issue, in that the clearest evidence supporting SR usage comes from unwieldy one-shot task designs that use a single-choice probe (following laborious training and retraining phases) to elicit slips of action that distinguish SR from full MB[20]. This makes it difficult to measure between-individual or -condition variation in SR usage, as well as tracking other more dynamic aspects of trial-by-trial SR learning and associated neural signals.

To study within-subject dynamics of SR usage, one promising approach is to examine its more granular, dynamic predictions about how past actions and outcomes guide future behavior on a trial-by-trial basis. This strategy has proven helpful in dissecting MB versus MF behavior, using two-step Markov decision tasks that expose differences in how MB vs. MF approaches update their choice preferences, trial by trial, in light of each trial's outcome[29]. Here, we pursued a parallel approach by designing a multi-step decision task that elicits unique trial-by-trial behavioral signatures of each planning strategy, reflecting recursive maximization for MB and on-policy state occupancy averaging for SR. Leveraging these signatures, we can characterize individual subjects' relative reliance on either approach, as a weighted mixture of the two strategies: estimating value via the multi-step state expectations derived from prior experience (SR), or a full tree search that maximizes at each step (MB).

The ability to characterize graded SR vs. MB usage within individuals allows us next to investigate whether people rationally adjust their reliance on the SR in response to circumstances such as environmental predictability. Although a line of theoretical and experimental work has focused on the judicious deployment of MB vs. MF[6,7], a recent line of theory suggests that a similar (and potentially more fundamental) trade-off may play out in the dynamic tuning of world-model abstraction between SR and MB[30,31]. Whereas the utility of MF learning versus MB planning depends on the stability of action value estimates[32], the predictive utility of the SR should analogously depend on the stability of long-run future state occupancy. We hypothesized that individuals would dynamically adjust their usage of the SR to the local temporal statistics of the task, and in particular that choices would increasingly reflect the SR when its state predictions were more stable and reliable. We estimated the balance of MB and SR contributions to behavior while manipulating these temporal statistics, and found that people rationally adjusted usage to match the accuracy of the SR. This suggests that humans can arbitrate between a temporally abstract model and full MB planning in response to their respective utility, adding to and recentering a growing literature on rational balance between decision mechanisms.

## Methods
### Participants
A total of 104 participants were recruited through Prolific to participate in an online behavioral experiment in September 2023. This study was approved by the Institutional Review Board of Princeton University, and all participants provided informed consent. The study was not preregistered. The total study duration was approximately forty minutes per participant. The participants received monetary compensation for their time (rate US $12 per hour), plus a task performance bonus up to $3.00 based on treasure collected in-game.

Participants were eligible if they were fluent in English, resided in Australia, Canada, New Zealand, or the United States, and had not participated in any prior versions or pilots of the study. Subject exclusion critera were implemented through Prolific. Participants self-reported gender as 51 male and 44 female. Two participants reported gender as "other" and three participants declined to report gender. Participants self-reported age, with ages between 18 and 68 years, mean = 36.19, SD = 11.55. 4 participants were excluded due to outlying response times, 18 participants returned the experiment on Prolific (no reason provided), and 2 participants dropped out due to technical issues.

**Code.** The task was implemented using NivTurk[33] and jsPsych[34] and distributed using custom web-application software. All experiment code is publicly available. A playable demo of the task is available at https://ariekahn.github.io/policy-sailing-task.

**Experimental task.** Participants performed a task where they repeatedly chose among four merchant boats, each with a hidden probability of reward, in order to maximize their monetary payout (a bonus of up to $3.00) by selecting the boats most likely to provide reward. The task was framed as a treasure-hunting game, consisting of 200 sets of alternating 'traversal' and 'non-traversal' trials. Participants were instructed that each merchant had a probability of reward independent of the other merchants, that these probabilities might change over the course of the game, that visiting a merchant did not increase or decrease future probability of reward, and that navigating to a merchant vs. having the same merchant visit the participant had the same probability of reward. The visual appearance of boats and islands was shuffled across subjects, as was the pattern of reward probabilities (see Reward Block Structure).

**Trial structure.** All trials started from the same 'home' island. On each 'traversal' trial, the game presented two candidate islands, and the participant chose to sail from their home island to either the left or right island by pressing the corresponding arrow on the keyboard. The game would then zoom in on the selected island, and two merchant boats would appear on the left or right side of the island, which incorporated a 1s delay, preventing the participant from entering their second-stage choice until this transition completed. The participant was now provided with a second choice between the two boats, and upon selecting a boat with the left or right arrow, it was revealed whether the chosen merchant would provide treasure - either a stack of gold bars (+1) or a red 'X' (+0) was shown on the screen. Notably, the position of the islands as well as the boats was constant throughout the experiment, though boats were not visible on the screen until the corresponding island was first chosen. On a 'non-traversal' trial, one of the four boats, at random, was shown to the participant at their home island. The participant pressed 'up' to select the boat and reveal whether the merchant would provide treasure. Each boat appeared 50 times, with a randomized order across the 200 'non-traversal' trials. Probability of reward was identical to the probability of reward for that boat on the preceding 'traversal' trial. In effect, these trials served to provide information about how likely one boat was to have reward, independent of the participant's own policy.

**Reward block structure.** The experiment consisted of 22 reward blocks of between 8 to 12 trial pairs. Reward probabilities were consistent within a given block, and systematically altered between blocks. At the start of the first block, one island consisted of merchants with probabilities of reward of 0.15 and 0.85, and the other island of merchants with probabilities of reward of 0.325 and 0.675. Assignment to each boat was randomized for each participant.

On a policy-consistent ('congruent') swap, the probabilities of the optimal boats on each island were swapped with one another, as were the probabilities of the sub-optimal boats. On a policy-inconsistent ('incongruent') swap, the probability of the optimal boat on one island was swapped with the probability of the sub-optimal boat on the other island. On a within-island swap (also 'incongruent'), the reward probability of each boat was swapped with that of the other boat on the same island. Suppose the boats are given by (A1, A2) on one island, and (B1, B2) on the other, with initial reward probabilities of (A1, A2, B1, B2) = [0.15, 0.85, 0.325, 0.675]. Then a policy-consistent swap would result in [0.325, 0.675, 0.15, 0.85], a policy-inconsistent swap would result in [0.625, 0.325, 0.85, 0.15], and a within-island swap would result in [0.85, 0.15, 0.675, 0.325]. These particular re-assignments were chosen such that in a congruent swap, a policy optimized for the previous reward structure will still correctly identify which island is better under the new rewards, but for an incongruent swap, a policy

optimized for the previous reward structure will *incorrectly* identify which island is better under the new reward structure.

**Data exclusion.** Participants whose mean reaction time was more than three standard deviations from the mean on either boat or island trials were excluded (1 outlier on island RT, 3 outliers on boat RT) leaving 100 subjects.

### Regression model
To test for a general effect of reward, behavior was fit to a hierarchical generalized linear mixed model using *MixedModels* v4.22.1[35] in *Julia* 1.9.3[36] with a Logit link function and the following formula:

$$actionTowardsSample \sim 1 + rwd \qquad (1)$$

where *actionTowardsSample* is whether the participant chooses the island associated with the sampled boat, and *rwd* is whether the sampled boat was rewarded. Per-subject random effects were included for both slope and intercept. For these error terms (and analogously for group-level variation in all subsequent models), we assumed normal distributions, but these were not formally tested.

To test for interactions representative of either policy-based or model-based credit assignment, behavior was fit to a hierarchical generalized linear mixed with a Logit link function and the following formula:

$$\begin{aligned} actionTowardsSample \sim{} & 1 + rwd + sibNoRwd + onPolicy \\ & + rwd \,\&\, sibNoRwd + rwd \,\&\, onPolicy \end{aligned} \qquad (2)$$

where *actionTowardsSample* is whether the participant chooses the island associated with the sampled boat, *rwd* is whether the sampled boat was rewarded, *sibNoRwd* is whether the most recently observed outcome from the second boat on the same island was unrewarded, and *onPolicy* is whether the previous action from the sampled boat's island was to the sampled boat. In addition, to control for non-independence between trials due to effects of events on previous trials, we added a number of lagged regressors: specifically, we included an additional four outcomes on the neighboring boat, four prior choices at the same island representing policy, the interactions of those with the reward of the previous trial, the previous five outcomes and interactions with reward of outcomes at the sampled boat, five lags of the average of the value of the two boats on the opposite island, and five lags of an island choice kernel for whether the participant had chosen the associated island on the last five traversals. All regressors are relative to the boat that was just sampled and the island on which that boat is found. Past values of the policy, current boat, and neighbor boat were coded 0/1, and all other regressors were coded $-0.5/0.5$.

### Behavioral choice model
To estimate the expected value of choosing each of the four boats, we used a simple value-learning algorithm, where the value of boat $i$ was updated after an observed reward $R_t$:

$$V(boat_i) \leftarrow (1 - \alpha)V(boat_i) + \alpha R_t \qquad (3)$$

The learning rate differed for traversal and non-traversal trials, to allow for separate active and passive learning rates: $\alpha \leftarrow \alpha_A$ on traversal trials and $\alpha \leftarrow \alpha_P$ on non-traversal trials, with both estimated independently of one another.

Choice between boats was modeled as a probabilistic decision between the two boats present on the chosen island on each trial using a softmax distribution:

$$P(b_t = b \in L, R) \propto \exp(\beta_{boat}V(b) + \beta_{persistenceB}LastChosen(b)) \qquad (4)$$

In human parameter estimation (but not simulation) $\beta_{persistenceB}$ was added to control for persistent choice behavior, where *LastChosen*(c) is 1 if $b$ was the most recently chosen boat at the current island, and 0 otherwise.

To estimate the expected value of choosing the left versus right island using a mixture of agents, values were estimated independently for a MB, SR, and TD(1) agent.

For the MB agent, the value of each island was simply taken as the maximum of the two estimated boat values:

$$V_{MB}(i) = \max(V(boat_1), V(boat_2)) \qquad (5)$$

For the SR agent, the value of each island was calculated via a matrix of future state occupancy, $M$, where $M$ was itself learned via a Hebbian learning rule, and $\alpha_M$ was a learning rate for the $M$ matrix which was fit as a free parameter:

$$M[island, boat] = \alpha_M 1_{boat=c} + (1 - \alpha_M)M[island, boat] \qquad (6)$$

Island values were then calculated as $V_{SR} = MR$, where $R$ is a vector of state rewards.

For the TD agent, island values were updated at the end of each traversal trial as follows:

$$V_{TD}(island) = (1 - \alpha_A)V_{TD}(island) + \alpha_A V_{TD}(boat) \qquad (7)$$

Choice between islands was modeled as a probabilistic decision between the two state values, again using a softmax distribution but with a separate inverse temperature for each agent ($\beta_{MB}$, $\beta_{SR}$, and $\beta_{TD}$), as well as persistence estimate $\beta_{persistenceI}$ in human model fitting:

$$\begin{aligned} P(i_t = i \in L, R) \propto{} & \exp(\beta_{MB}V_{MB}(i) + \beta_{SR}V_{SR}(i) \\ & + \beta_{TD}V_{TD}(i) + \beta_{persistenceI}LastChosen(i)) \end{aligned} \qquad (8)$$

### Linear RL choice model
To estimate the expected value of choosing the left versus right island using linear RL, values were estimated from a transition matrix $\mathbf{T}$ and rewards $\mathbf{r}$ (both learned over the course of the experiment):

$$\exp(\mathbf{v}_N^*/\lambda) = \mathbf{MT}_{NT}\exp(\mathbf{r}_T/\lambda) \qquad (9)$$

where $\mathbf{v}_N^*$ is a vector of optimal values at non-terminal states, $\mathbf{M}$ is the *default representation* (DM) matrix from all non-terminal to all other non-terminal states, $\mathbf{T}_{NT}$ is the transition matrix from all non-terminal to terminal states, $\mathbf{r}_T$ is the set of rewards (or negative step costs) at all terminal states, and $\lambda$ arbitrates between exact and policy-guided estimates. Value estimates of the boats were given by taking the logarithm of the left-hand side and multiplying each value by $\lambda$.

The DR matrix is in turn given by

$$\mathbf{M} = (\text{diag}(\exp(-\mathbf{r}_N/\lambda)) - \mathbf{T}_{NN})^{-1} \qquad (10)$$

where $T_{NN}$ is the transition matrix from all non-terminal to all other non-terminal states and and $\mathbf{r}_N$ is the set of rewards at all non-terminal states.

$r$ was assumed to be 0 at islands, and for boats equal to the estimated $V$ for that boat.

$T$ was updated after each choice using a Hebbian learning rule, which was chosen to provide equivalent updates to those of the SR's $M$ matrix:

$$T[island, boat] = \alpha_T 1_{boat=c} + (1 - \alpha_T)T[island, boat] \qquad (11)$$

Choice between islands was modeled as a probabilistic decision between the two state values estimated via linear RL, again using a softmax distribution but with a separate inverse temperature $\beta_{island}$, as well as persistence estimate

**Fig. 1 | Planning model and task structure. a** Trials alternate between "traversal" trials (top) where subjects choose an island and then a boat, and "non-traversal" trials (bottom) in which a boat is selected at random and its reward delivered, without island choice or presentation. On each traversal trial, the participant selects either the left or right island. Upon selecting an island, the two boats available on that island appear, and the participant selects either the left or right boat. On each non-traversal trial, the participant is not given the option to select an island or boat, and instead, one of the four boats "visits them" at the starting location, with identical payout probabilities. The locations of all boats are fixed for the duration of the task. Bottom Right: Full schematic of trial structure. (Art adapted from Vecteezy.com). **b** A MB agent (left) evaluates choices via recursive maximization, while an SR agent (center) evaluates choices via on-policy averaging. Both SR and MB agents are capable of integrating nonlocal information through the use of an appropriate world model. A MF agent (right) directly learns the values of actions and thus cannot integrate nonlocal information. **c** The experiment consisted of 22 reward blocks of between 16 and 24 trials, alternating between traversal and non-traversal trials. Reward probabilities were consistent within a given block, and systematically altered between blocks. Image credits: islands (graphicsrf/Vecteezy), boats (icon0/Vecteezy).

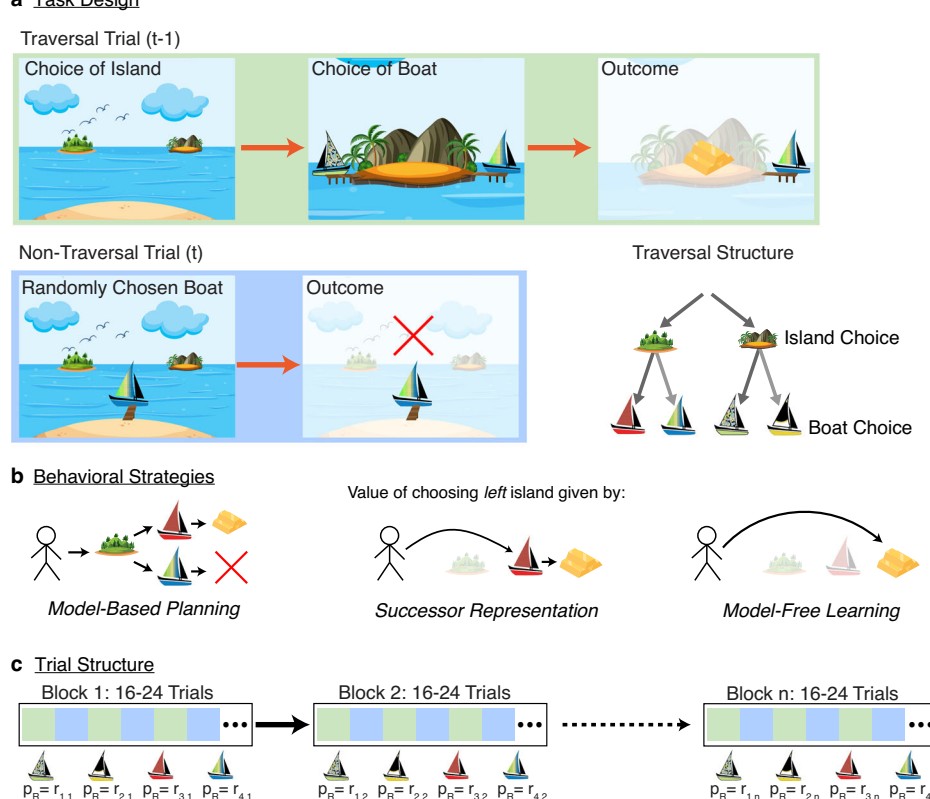

$\beta_{persistenceI}$ in human model fitting:

$$P(i_t = i \in L, R) \propto \exp(\beta_{island}V(i) + \beta_{persistenceI}LastChosen(i)) \quad (12)$$

## Parameter estimation

We optimized the free parameters of the learning algorithms by embedding each of them within a hierarchical model to allow parameters to vary between subjects or simulated sessions. Subject-level parameters were modeled as arising from a population-level Gaussian distribution over subjects. We estimated the model to obtain best fitting subject- and group-level parameters to minimize the negative log likelihood of the data using an expectation-maximization algorithm with a Laplace approximation to the session-level marginal likelihoods in the M-step[37]. For hypothesis testing on population-level parameters, we computed an estimate of the information matrix over the population-level parameters, taking account of the so-called "missing information" due to optimization in the E-step[38], itself approximated using the Hessian of a single Newton-Raphson step.

## Blockwise change analysis

To estimate whether subjects shifted their strategies after congruent versus incongruent block changes, we fit a two-step choice model parameterized in terms of $\beta_{TD}+$, $w_{SR}+$, and $\beta_{MBSR}+$ (used following a congruent block change), and $\beta_{TD}-$, $w_{SR}-$, and $\beta_{MBSR}-$ (used following an incongruent block change), where $\beta_{MB}+ = (1-w_{SR}+)\beta_{MBSR}+$, $\beta_{SR}+ = (w_{SR}+)\beta_{MBSR}+$, $\beta_{MB}- = (1-w_{SR}-)\beta_{MBSR}-$, and $\beta_{SR}- = (w_{SR}-)\beta_{MBSR}-$. This change of variables (analogous to one often used for MB vs. MF trade-offs; Daw et al.[29]) was chosen to allow straightforward comparison and statistical testing between $w_{SR}+$ and $w_{SR}-$. Note that again there is no inherent constraint that the congruent block values should be greater or less than the incongruent block values.

For the linear RL model, island values were calculated with linear RL (see above) but with two independent values of $\lambda$, termed $\lambda+$ and $\lambda-$, and two softmax temperatures, $\beta_{LRL}+$ and $\beta_{LRL}-$. $\lambda+$ and $\beta_{LRL}+$ were used to estimate island values throughout a block following a congruent reward change, whereas $\lambda-$ and $\beta_{LRL}-$ were used to estimate island values throughout a block following an incongruent block change, in addition to $\beta_{TD}+$ and $\beta_{TD}-$. Note that again there is inherent constraint that $\lambda+$ should be greater or less than $\lambda-$.

## Reporting summary

Further information on research design is available in the Nature Portfolio Reporting Summary linked to this article.

## Results
### Task

Participants (n = 100) completed a two-stage Markov decision task in which they repeatedly chose between two islands and then between one of two pairs of boats, each boat with an unsignaled probability of reward. Their goal was to maximize their total monetary payout by selecting the boats (and, in turn, islands) most likely to provide treasure. The task was framed as a treasure-hunting game, consisting of 200 pairs of alternating 'traversal' and 'non-traversal' trials. Traversal trials allowed subjects to navigate to one of two islands, and then one of two boats (Fig. 1a, top), whereas non-traversal trials involved no decision making, with one of the boats visited at random (Fig. 1a, bottom left). Thus, non-traversal trials provided information about the value of boats, but without explicitly associating that information with the path to arrive at that boat. Importantly (compared, for instance, to other two-step Markov decision tasks), this ensured that any effect of this information on the next island choice was mediated via some type of a world model (i.e., it ruled out MF learning about the island choice on these trials), enabling a direct comparison of MB vs. SR approaches to this type of model-based credit assignment (Fig. 1b).

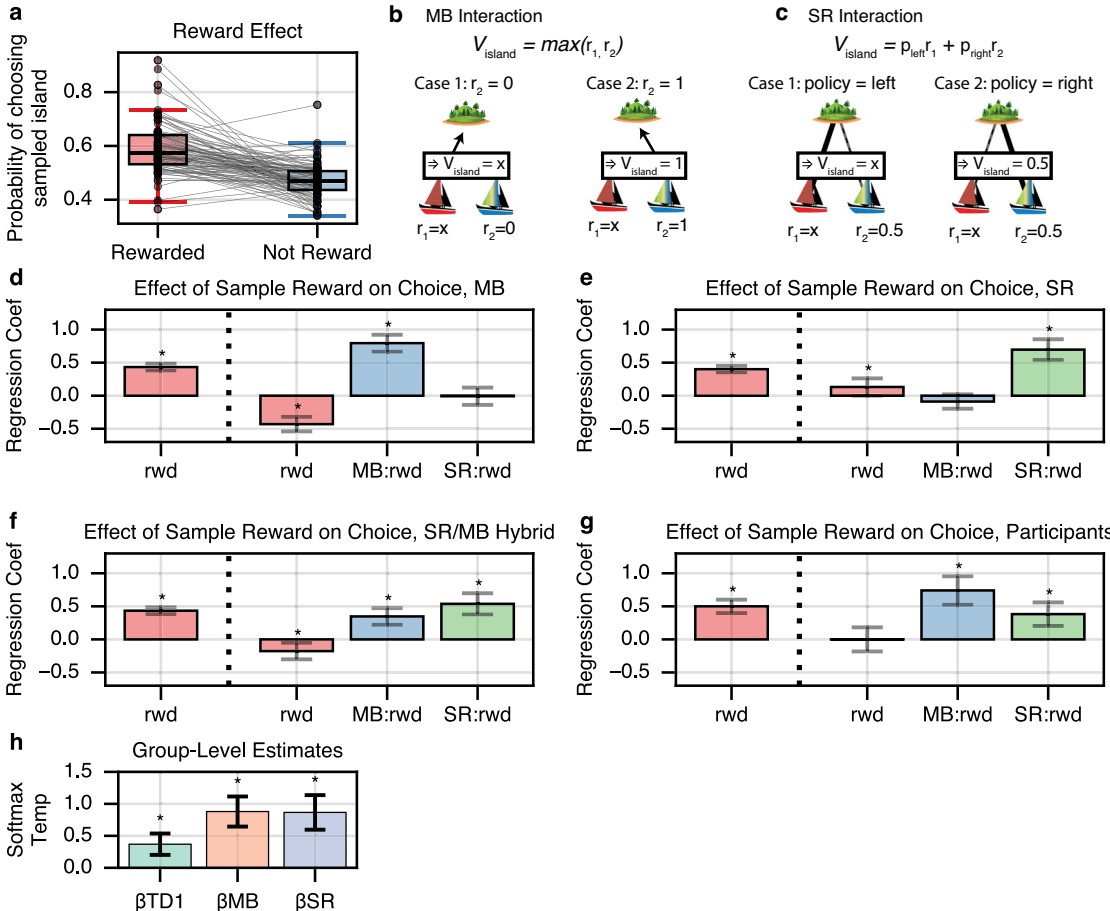

**Fig. 2 | Distinct signatures of MB and SR. a** Reward Sensitivity: Participants are significantly more likely to choose the island associated with a sampled boat when the boat was rewarded, behavior which cannot be explained by MF planning mechanisms. Each line is one subject. **b** MB signature: A MB agent maximizes over boat values at an island, so it is only sensitive to a boat's reward (or nonreward), here indicated as 'x', if the neighboring boat has low value (recently unrewarded). **c** SR signature: An SR agent predicts value from expected boat occupancy, which depends on previous choices at the island and is thus only sensitive to a boat's reward if that boat was chosen previously. **d–g** Effect of non-traversal reward on subsequent

traversal island choice. Left-most bar: Reward-only model. All four models show a significant effect of modulation due to reward on the previous trial. Right three bars: Model including MB and SR interaction terms. Effect of reward almost entirely disappears when interactions with neighboring boat and prior policy are included. MB learners show a strong interaction with the neighboring boat, whereas SR learners show a strong interaction with prior choice at the sampled island. Hybrid agents, as well as humans, exhibit both interaction effects. **h** Estimated group-level softmax temperatures for TD(1), MB and SR strategies. Error bars are 95% CI. Asterisks indicate $p < 0.05$.

Participants were instructed that reward probabilities were properties of the boats alone, independent from one another and from the particular action sequences the subjects might take to achieve them (including being the same on traversal vs. non-traversal trials), but might change over the course of the game. Nonetheless, rewards did follow certain patterns, unknown to the subjects: reward probabilities were fixed for blocks of 16–24 trials (or 8–12 pairs of traversal and non-traversal trials), after which they would switch to a new set of reward probabilities for the subsequent block (Fig. 1c). Of note, the probabilities were structured so that at all times both islands had an equal probability of reward, if boats were chosen at random, so that non-traversal trials were equally likely to result in reward attributable to either island.

**Participants use world models to incorporate reward information**
The task was designed to distinguish between MB and SR strategies on a per-trial basis, because these predict different characteristic patterns by which reward information from non-traversal trials would be incorporated into subsequent decisions. We can understand and provide an initial test of the models' predictions by focusing specifically on the effect of each non-traversal trial's reward on the island choice made on the subsequent traversal trial. These types of simplified, one-trial-back analyses help complement

and provide intuition about the factors driving the relative fit of more elaborate learning models, which we consider later. First, to verify that participants were indeed incorporating reward information from these trials, we examined whether receiving reward (vs. not) for some boat on a non-traversal trial affected participants' chance of choosing the island associated with the sampled boat on the immediately following traversal trial. Both approaches to world modeling (MB and SR) predict a win-stay-lose-shift pattern, on average, at the island level: that is, to the extent they follow either strategy, participants should be more likely to choose the associated island if the boat was rewarded rather than unrewarded on the immediately succeeding trial.

Indeed, participants were more likely to choose the associated island when the previous non-traversal trial was rewarded than when it was unrewarded (effect of reward: $\beta = 0.5$, $z = 9.66$, two-sided $p < 1e-21$; mixed effects logistic regression), indicating a baseline sensitivity to reward (Fig. 2a). (Note that this effect cannot be ascribed to subjects being in any way more likely to prefer islands associated with rewarded boats, since the probability the sampled boat is rewarded is at all times matched across islands.) Importantly, since the choice of an island was not experienced on a non-traversal trial, this effect implies the use of some type of internal model to infer the implication of a boat's reward for the choice of islands. This

allowed us to further decompose this reward effect, to investigate to what extent it reflected the contributions of either SR or MB planning mechanisms.

We next fit the same pair of models to human participants (Fig. 2g), and find that similar to a mixture of MB and SR planning, participants exhibited a sensitivity to reward from non-traversal trials ($\beta = 0.5$, 95% CI = $+/- 0.101$, $p < 4.61e-22$) which was entirely mediated by the two interaction terms ($\beta = 0.0002$, 95% CI = $+/- 0.182$, $p < 0.998$), whereas both interactions with reward show significant positive effects (MB effect: $\beta = 0.743$, 95% CI = $+/- 0.215$, $p < 1.32e-11$, SR effect: $\beta = 0.384$, 95% CI = $+/- 0.179$, $p < 2.53e-5$). Notably, this mixture of MB and SR behavior is in-line with previous work such as Momennejad et al.[20], which also suggested (using a much different task and blockwise manipulation) that human behavior could be explained with a mixture of the SR and MB planning, but not with either individually. Likewise, we observed in our task that human behavior, at the group level, displayed a similar mix of behaviors: people rely on both SR and MB evaluations.

## Mixture of agents model

While the above model-agnostic analysis helps to provide evidence and intuition for the separability of MB and SR behavioral signatures, actual subject choices depend on expectations learned incrementally over multiple trials. Thus to complement this model-agnostic analysis and further verify that human choice behavior was best-explained by a mixture of MB and SR planning, we fit subject behavior with a fuller learning model, extending the hybrid "mixture of agents" model often used previously in sequential decision tasks[29]. Previous models typically contained two agents: MB alongside a model free (temporal difference, TD, learning) agent. Here we added to this mix a third, an SR agent. Note that since the MF agent learns experientially, it cannot directly account for reward effects on non-traversal trials; but it might still contribute to choice via preferences learned on traversal trials, whose influence its inclusion accounts for. The three agents' contributions to choice are weighted by separate softmax temperatures ($\beta_{MB}$, $\beta_{SR}$, $\beta_{TD}$), which measure their respective contribution to choice.

The agents shared learning rate parameters. In particular, previous work suggests that people may employ different learning rates for active versus passive learning[39,40], and so we allowed for learning rate to vary between the two trial types: $\alpha_A$ on traversal trials, and $\alpha_P$ on non-traversal trials (active and passively, respectively).

Fits of this model indicated human choice behavior exhibited significant signatures of all three agents, with group-level estimates of the softmax temperatures of all three agents significantly different from zero (Fig. 2h) ($\beta_{TD} = 0.369$, 95% CI = $+/- 0.167$, $t_{810} = 4.328$, $p < 1e-4$, $\beta_{MB} = 0.88$, 95% CI = $+/- 0.234$, $t_{810} = 7.33$, $p < 1e-10$, $\beta_{SR} = 0.866$, 95% CI = $+/- 0.27$, $t_{810} = 6.282$, $p < 1e-9$). Consistent with the model-agnostic regressions, this result indicates that human behavior exhibits distinct characteristics of MB and SR planning, even with additional MF contributions taken into account, and that choice behavior is best-fit by a model that allows for a mixture of the three.

## Within-subject arbitration between SR and MB strategies

Given the signatures of both SR and MB evaluation, we next wished to investigate how participants balanced these strategies. In particular, extending previous ideas about MB vs. MF arbitration[6–8], we hypothesized that participants should tend to adopt an SR strategy when the simplification underlying it was more often valid, and fall back on more expensive MB computation when the SR's assumptions were more often violated. In particular, the SR simplifies iterative prediction by assuming that expectations about state encounters (here, boats given islands) are consistent over time: in our task, this requires that participants' own per-island boat preferences are stable. To test this, our task manipulated the validity of this strategy through systematic blockwise switches in reward probability which induced more or less policy instability. The rewards associated with each of the four boats underwent frequent switches (Fig. 3a).

In particular, at the beginning of the experiment, reward probabilities were initialized such that each island had a more and a less rewarding boat, and the better boat in one island was in turn better than the other island's (better $p(reward) = 0.85$ vs. $0.675$). Every 16–24 trials, reward probabilities would reverse in one of two ways, either favoring or disfavoring the SR. First, the rewards might undergo a *congruent change*, (15/21 block changes), in which probabilities were swapped between the two higher boats and between the two lower boats, such that optimal choice of island changed, but optimal choice of boat conditioned on island remained the same (Fig. 3a, left). Second, rewards might undergo an *incongruent change, 6/21 block changes*, which changed the optimal boat choice at each island (Fig. 3a, right). (Half of these also swapped the optimal island choice; half did not). The key observation is that the incongruent block changes violate the assumptions of the SR: island choices based on the SR will mispredict the island's optimal value (recommending suboptimal island choices), and to the extent participants relearn new boat choices, the SR's predictions about boat encounters will initially be violated, and the SR will require updating. We hypothesized that the brain could detect these effects (e.g., via prediction errors on boat encounters) and adaptively reduce SR usage following incongruent block shifts. (We included fewer incongruent than congruent reversals based on the expectation, confirmed in piloting, that selective adaptation would be most likely elicited by occasional rather than frequent challenges).

To estimate whether subjects' balance of MB and SR planning systematically varied across blocks, we again fit the mixture of agents model to choice behavior, but now estimating separate MB, SR, and TD(1) softmax temperatures for each block type, determined by which type of reversal began that block. We predicted that when subjects encountered new reward probabilities under which optimal choice of island was consistent with prior policy, participants would rely more heavily on their prior policy, and their behavior would be best estimated with a larger relative reliance on SR over MB, reflected in higher SR and/or lower MB softmax temperatures. If, instead, new reward values led to incorrect choice of islands under the prior policy, we expected that participants would spend relatively more effort on explicit MB planning, which would be reflected in increased MB and/or decreased SR softmax temperatures. To formally test for blockwise changes in relative SR vs. MB reliance, we computed their fractional contribution $w_{SR} = \beta_{SR}/(\beta_{SR} + \beta_{MB})$, such that a weight of 0 indicated complete reliance on MB and a weight of 1 complete reliance on SR[29]. (This was accomplished by estimating the original model using a change of variables; see Methods).

Estimating the model hierarchically across subjects, we found that the group-level mean $w_{SR}+$ (the reliance on SR relative to MB planning following a congruent reward change) was, as hypothesized, significantly larger than $w_{SR}-$ (the value following an incongruent reward change). Following a congruent reward change, $w_{SR}+$ was estimated at 0.604 (95% CI: [0.496, 0.705]) indicating an SR-dominant strategy. Following an incongruent reward change, $w_{SR}-$ was estimated at 0.336 (95% CI: [0.197, 0.504]), indicating a shift to an MB-dominant strategy. This change was itself statistically significant (estimated difference in $\log(w_{SR}) = 0.687$, $t_{1044} = 2.797$, $p < 0.006$, 95% CI = [0.205, 1.169]). In contrast, there was no significant blockwise difference in the MF contribution, ($\beta_{TD}+ = 0.236$, 95% CI = [0.066, 0.406], $\beta_{TD}- = 0.156$, 95% CI = [−0.018, 0.330], estimated difference = 0.079, 95% CI = [−0.136, 0.294], $t_{1044} = 0.720$, $p < 0.47$), nor in the overall MB/SR softmax temperature $\beta_{MBSR}$ that, on this change of variables, measures the contribution of a $w_{SR}$-weighted average of MB and SR values ($\beta_{MBSR}+ = 2.029$, 95% CI = [1.669, 2.388], $\beta_{MBSR}- = 1.637$, 95% CI = [1.274, 2.001], estimated difference = 0.392, 95% CI = [−0.075, 0.859], $t_{1044} = 1.644$, $p < 0.11$).

These results indicate that on blocks following congruent reward changes, subjects exhibited increased reliance on temporally abstract planning, whereas following incongruent reward changes, subjects turned more to MB evaluation. This difference was also relatively consistent in the hierarchical model's subject-level estimates: 93/100 participants were individually estimated with $w_{SR}+ > w_{SR}-$. Thus we observe that people are

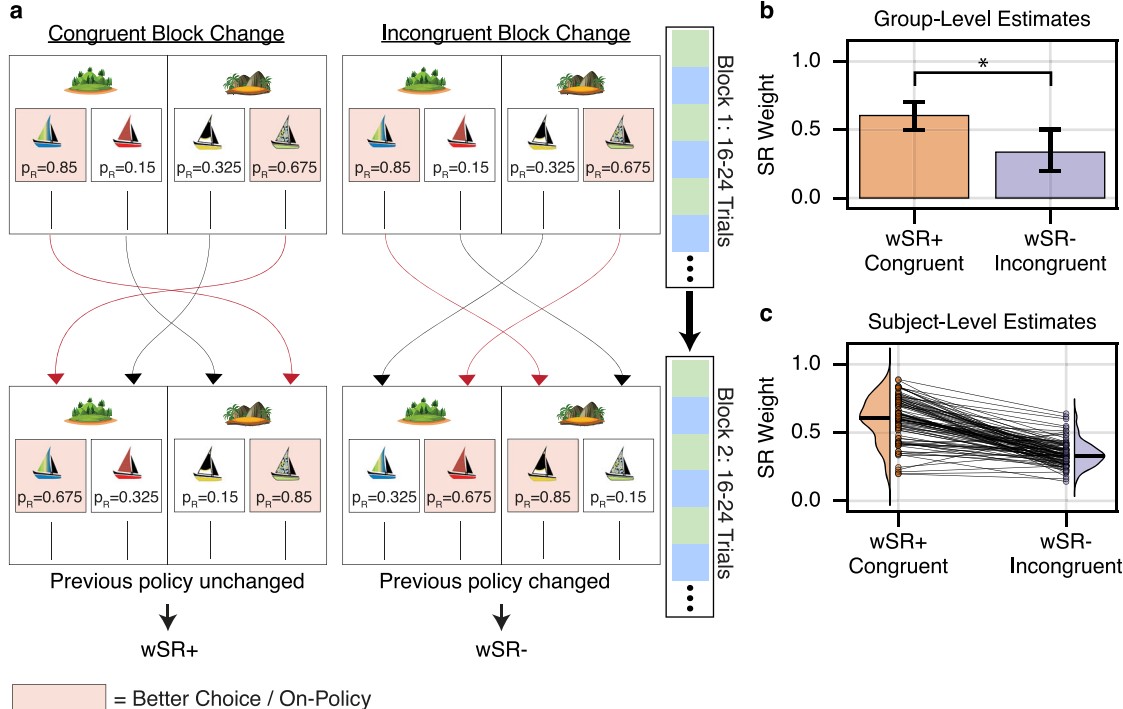

**Fig. 3 | MB/SR balance systematically varies across block reward structure. a** Left: Congruent block change. After 16-24 trials, reward probabilities swapped such that the optimal choice of island flipped, but the optimal boat choice conditioned on island choice remained unchanged. Crucially, evaluating the new rewards via the policy induced by previous rewards still provided the optimal choice of island. Right: Incongruent block change. Again, the optimal island changed, but now, the policy induced by previous rewards valued the sub-optimal island more highly. **b** Parameter fitting results. A hierarchical choice model was fit across subjects which included a weighting term $w_{SR}$ parameters controlling the balance between MB and SR behavior. $w_{SR}+$ was used on blocks following congruent reward changes, whereas $w_{SR}-$ was used on blocks following incongruent reward changes. $w_{SR}+$ was significantly greater than $w_{SR}-$ at the group level (estimated difference in $\log(w_{SR}) = 0.687$, two-sided $t_{1044} = 2.797$, $p < 0.006$, 95% CI = [0.205, 1.169]), suggesting that subjects increasingly relied on the SR when future state occupancy was stable, and MB planning otherwise. Error bars show $+/-$ 1.96 SE. **c** Individual subject fits. Each line is one subject.

capable of reacting to the temporal predictability of their current environment, and dynamically adjusting their decision-making strategy to balance cost-of-planning against expected outcome.

We further confirmed that this difference in estimated parameters was not induced by the blockwise structure of the experiment itself: when simulating an agent insensitive to the blockwise structure and recovering softmax temperatures for each block, we found that there was not a significant estimated shift between block types (Supplementary Fig. 2).

### Linear RL

Although we have so far estimated SR and MB contributions via separate modules in a standard mixture-of-agents model, recent theoretical work suggests an alternative formulation in which both algorithms arise as special cases of a single world model computation with tuneable reliance on temporal abstraction[30,41]. Though largely behaviorally equivalent, this suggests a different view on the neural implementation of world models and the cost of MB planning.

In particular, a linear RL agent computes values under a default assumed dynamics (equivalent to SR's $M$), but can partly correct for this and approximate the iterative maximization underlying MB choice using a softmax-like nonlinearity[30,41]. The degree of this correction is controlled by a single scale factor, $\lambda$, which, like $w_{SR}$, parameterizes a spectrum of behavior from SR to MB (Fig. 4a). Furthermore, this parameterized scaling implies a concrete computational cost in terms of bits of precision (e.g. more spikes), as better approximations to MB choice require computing larger, more precise values: (Fig. 4b). As expected (since the model is largely behaviorally equivalent on the current task), this alternative specification, when estimated on our data, recapitulated our core results. In particular $\lambda$ was overall estimated as intermediate (between extreme values characteristic of pure SR

or MB) (Fig. 4c), and declined (nearer to MB) on incongruent blocks when compared to congruent blocks (Fig. 4d, e).

### Discussion

When do we spend time and mental resources iteratively simulating the future, and when do we rely on simpler procedures? Simplified evaluation strategies inherently trade off planning complexity against accuracy, and the usefulness of any given approach for planning depends on the environment an agent is operating within[6]. This has often been discussed in terms of a trade-off between model-based and model-free methods. MF learning is advantageous when the long-run values of candidate actions change slowly enough for a cached estimate to accurately reflect distant reward outcomes[32]. It has been hypothesized that the brain trades off such strategies judiciously according to circumstance[42], and (in the case of MB vs. MF arbitration) that this process explains phenomena of both healthy automaticity and pathological compulsion[6,12,43,44].

Here we extended this program to the question of how the world model itself works. Simplified world models, such as the SR, have seen recent attention in decision neuroscience. These simplify planning not by foregoing model-based simulation altogether, but instead by collapsing the iterative simulation of dynamics by using temporal abstraction over future state encounters: in effect, adopting a simplified, multi-step world model[18,19,30]. Furthermore, one candidate implementation of full MB planning in the brain involves an inherent, tuneably cost-saving trade-off with SR-like abstraction, suggesting that judicious MB-SR trade-offs may be as, or even more, fundamental than the classic MB-MF picture[30]. The SR and related methods provide a potential solution to how the brain can plan efficiently over temporally extended timescales, so long as the associated predictions are stable, but also predict subtler biases. We introduced a new

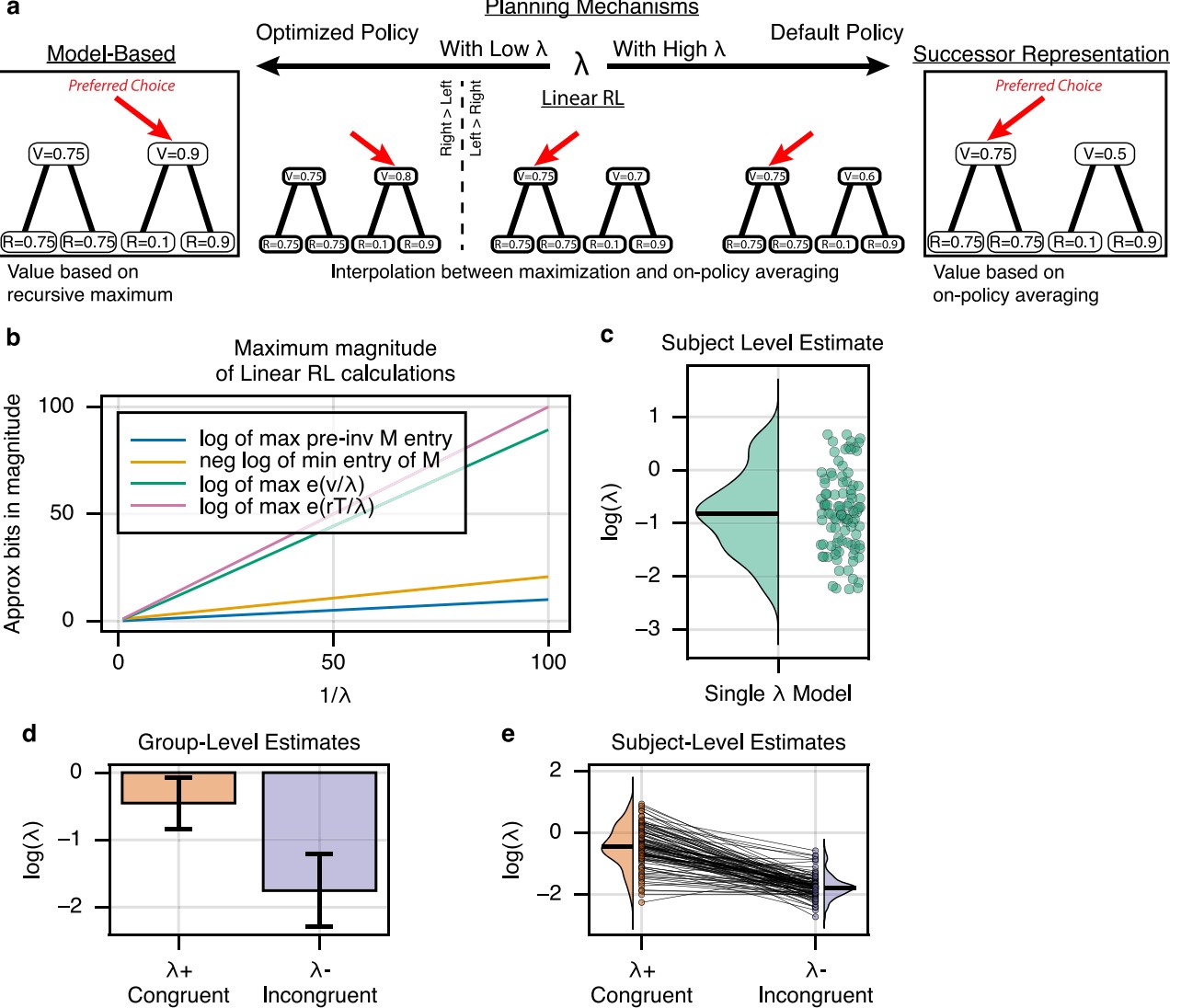

**Fig. 4 | Linear RL λ systematically varies across block reward structure. a** Linear RL encapsulates both MB and SR behavior. As a single parameter λ varies between extremes, linear RL approaches either MB (low λ) or SR (high λ) evaluation. **b** Tuning for MB behavior exhibits an inherent cost. As λ approaches zero, the necessary computational precision dramatically increases, suggesting a fundamental neural cost to increasingly accurate read-outs. Here, the relationship between the magnitude of the largest value and the inverse of λ is shown for the current task. **c** Parameter fitting results. A hierarchical choice model was fit across subjects which included a single λ parameter controlling the balance between MB and SR behavior. $log(λ)$ was estimated at a value of $-0.803$, 95% CI $= +/- 0.323$, suggesting that

subjects rely on a mixture of MB and SR-like behavior (Supplementary Fig. 1). Each point is one subject. **d** Blockwise fits. A hierarchical choice model was fit across subjects which included two λ parameters controlling the balance between MB and SR behavior. λ+ was used on blocks following congruent reward changes, whereas λ− was used on blocks following incongruent reward changes. $log(λ+)$ was significantly greater than $log(λ−)$ at the group level (difference = 1.301, two-sided $t_{728} = 3.947$, $p < 8.5e−5$, 95% CI $= [0.655, 1.947]$), suggesting that subjects increasingly relied on the SR when future state occupancy was stable, and MB planning when unstable. Error bars show $+/- 1.96$ SE. **e** Individual subject fits. Each line is one subject.

task and analyses that allowed us to track SR vs. MB usage dynamically in terms of trial-by-trial choice adjustments rather than, as in previous work, via one-shot probes[20]. Using these methods, we confirmed previous suggestions that people use such temporally abstract models (the SR) alongside iterative MB planning mechanisms. This allowed us further to investigate dynamic strategy usage within-participant, providing evidence that people dynamically adjust the balance between these strategies in response to their accuracy.

**Dynamic adaptation**

Previous research on arbitration between MF and MB planning systems suggests that people dynamically adjust their behavioral strategies, optimizing for a cost-benefit trade-off[7,8] between fast but noisy estimates and slow deliberation. For example, when faced with frequently shifting long-

run outcomes, people should increasing rely on MB strategies, despite the additional cognitive cost[32]. That is, the brain is capable of recognizing when cached value estimates are likely to be unreliable, and spends additional time and effort to compensate. Here, we showed that the same is empirically true of SR and MB planning: when cached policy estimates were unreliable vis-a-vis optimal rewards, people shifted towards MB strategies.

What causes a shift in reliance on the SR versus MB planning? Prior work on MF behavior suggests that people may rely either on prediction accuracy[6], or on estimates of sub-optimality of reward obtained under the current policy[8]. Here, our experimental design exhibited both of these characteristics at the block level: incongruent blocks led to both more frequent state prediction errors, as well as increased losses if islands were chosen via the previous policy. Yet another possible signal is whether outcomes are generally consistent with the future occupancy expected under

the SR — if rewards tend to occur in states on which we put little future expected weight, or vice-versa, we may infer that the current policy is invalid. While a long-term solution is to relearn updated state occupancy estimates, a short-term reaction may be to temporarily reduce reliance on the SR, and instead perform more costly MB searches. The reliance on a blockwise measure of arbitration in the current study allows us to capture coarse behavioral shifts, but future work will be required to test which precise characteristics drive these shifts, including the use of finer-grained models which these these hypothesized mechanisms of trial-by-trial adaptation against the current behavioral paradigm.

### Trial-by-trial dynamics of temporal abstraction

Our work builds on the logic of previous approaches to experimentally distinguish SR-based from step-by-step MB planning. One key signature of the SR is a limited ability to successfully replan when distant rewards change: replanning is hampered (and step-by-step MB re-evaluation required) when reward changes also affect the preferred choice policy at subsequent states[19]. To test for this limitation, previous approaches have relied on one-shot train-retrain-probe designs ("policy revaluation" retraining), contrasting one-shot change scenarios in which an SR planner should vs. should not be able to successfully replan. Because of their binary nature (and susceptibility to practice effects if repeated), these provide limited information about individual differences, instead only providing a generalized summary of human behavior on average over subjects[20]. Relatedly, this class of designs is also not ideal for extended training in animals, nor for dynamic neuroscientific measurements and manipulations, like neuroimaging, electrophysiology, or optogenetics.

Through design of a choice task, we extend the logic of policy revaluation to the per-trial effects of individual rewards on trial-by-trial choice adjustment. Specifically, the reliance of MB planning on successive maximization and of SR on the previously experienced policy both lead to distinct effects of updated reward information, when that reward information is interspersed between trials probing the subject's action preferences. In effect, in this task each "non-traversal" trial is a miniature version of a (policy and/or reward) revaluation training phase. By measuring these effects we can measure the influence of these interactions over the course of the experiment. Extending previous approaches that used similar logic and designs to distinguish MB from MF updating[29], we show that trial-by-trial learning dynamics provide a rich source of information about the details of internal planning computations: in effect, how participants model the world's dynamics for the purpose of evaluating choices.

Indeed, similar to the large body of previous work using the two-step Markov decision task to distinguish MB from MF, we expect that the ability to measure SR usage dynamically will analogously enable informative future studies. These might include the ability to measure and manipulate neural correlates of these planning processes dynamically during learning, to further investigate arbitration driven by cost-benefit trade-offs between multiple planning systems[7,8], and, via better characterization of individual differences in these behaviors, to investigate whether usage of SR systematically varies in psychiatric conditions (much as has been argued for MF)[12], as well as over the course of normal human development[15].

### Trade-offs in linear RL

Traditionally, phenomena of automaticity vs. deliberation have been viewed in terms of MF vs. MB competition: retrieving cached long-run values vs. updating them via model search[6]. The SR, in turn, was first conceived as a third, intermediate strategy, which flattens and thus more cheaply approximates model search by caching long-run trajectories[18,19]. This is indeed the structure of the descriptive, mixture-of-agents model by which we primarily analyze our data.

However, recent theoretical work on linear RL[30] suggests an alternative implementation of MB computation in the brain, in which a trade-off against SR-like simplification occurs intrinsically within a single computation. In linear RL, values are predicted under an assumed multistep

dynamics, similar to the SR, but with an additional softmax-like nonlinearity that can, in the limit, mimic full MB computation. This trade-off, in turn, has a concrete cost (e.g. in terms of spikes or time): more MB-like values require larger, more precise decision variables. On this view, then, tuneable MB vs. SR trade-offs are inherent to word modeling, and potentially as fundamental as MB vs. MF. This is thus a natural framework for considering rational meta-control of the sort we find evidence for here.

### Limitations

The suggestion that the range of behaviors studied here might be realized, neurally, by a single linear RL agent points to one key limitation of the current study: the difficulty distinguishing evaluation mechanisms from choices alone. Indeed, the current experiment was not designed to distinguish linear RL from a mixture of separate SR and MB agents – both of which suffice, at a descriptive level, to capture our key findings. Thus, our results do not yet speak directly to whether the underlying operations, and their evidently rational balancing, are implemented in a single or separate neural "modules." This might most plausibly be examined, in future studies, using neural measurements or manipulations.

The ambiguity between the mixture of agents vs. the linear RL models points to a more general challenge in research of this sort, which is the possibility that additional unanticipated model variants might also (or even better) explain the results, compared to the models hypothesized here. In general, our model-driven analyses make relatively strong assumptions about the parametric forms of the computations, which are intended to be broadly representative of key computational approaches. This allows us to control formally for potential confounds like additional contributions of MF, but may leave open the possibility that (for instance) some other MF variant might still apply. To complement this approach, we conduct additional analyses with a more general class of models (here, linear regression in lagged events), which imposes different and arguably weaker assumptions but is also, perhaps, statistically less powerful. One frontier of ongoing research is to better address this trade-off by combining more flexible data-driven model discovery (e.g. using deep networks) fed by much larger datasets.

### Data availability

Data that support this study are available at https://osf.io/ncq34/as well as https://github.com/ariekahn/sailing-paper-code.

### Code availability

The experiment code is available at https://github.com/ariekahn/policy-sailing-task. The custom web software for serving online experiments is available at https://github.com/nivlab/nivturk. A playable demo of the task is available at https://ariekahn.github.io/policy-sailing-task. All analysis code is available at https://github.com/ariekahn/sailing-paper-code.

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

## Acknowledgements

This work was supported by grant 61454 from the John Templeton Foundation, grant W911NF-16-1-0474 from the U.S. Army Research Office, and and grant R01MH121093 from National Institute of Mental Health. The funders had no role in study design, data collection and analysis, decision to publish or preparation of the manuscript. The content is solely the responsibility of the authors and does not necessarily represent the official views of any of the funding agencies. We would like to thank Rani Moran and Kate Nussenbaum for helpful feedback on a previous version of this manuscript.

## Author contributions

A.E.K. and N.D.D. conceived the project and planned the experiments and analyses. A.E.K. performed the experiments and analyses. A.E.K. and N.D.D. wrote the manuscript.

## Competing interests

The authors declare no competing interests.
