## [Peer Review File · Communications Psychology]

This manuscript has been previously reviewed at another *Nature Portfolio* journal. This document only contains reviewer comments and rebuttal letters for versions considered at *Communications Psychology*.

12th Aug 24

Dear Nathaniel,

Your manuscript titled "Humans rationally balance detailed and temporally abstract world models" has now been seen by two of the reviewers who vetted the earlier version. Their comments appear below. In light of their advice I am delighted to say that we are happy, in principle, to publish a suitably revised version in Communications Psychology.

We therefore invite you to revise your paper one last time to address the remaining concerns of our reviewers and a list of editorial requests. At the same time we ask that you edit your manuscript to comply with our format requirements and to maximise the accessibility and therefore the impact of your work.

EDITORIAL REQUESTS:

SUBMISSION INFORMATION:

OPEN ACCESS:

* **DATA AVAILABILITY:**

[link redacted]

Best wishes,

Marike

Marike Schiffer, PhD

Chief Editor

Communications Psychology

REVIEWERS' COMMENTS:

Reviewer #1 (Remarks to the Author):

The authors have significantly improved the manuscript, making it much clearer and more accessible. My concerns, particularly regarding MF contributions to behavior and the differential learning rate for traversal and non-traversal trials, have been satisfactorily addressed.

I am pleased to recommend the publication of this manuscript.

Reviewer #2 (Remarks to the Author):

I commend the authors on a thorough and judicious revision. I think the new balance between the mixture model and the forward-looking discussion of LRL is just right. I have carefully reviewed the revised manuscript and I recommend publication in its current form.